# Real World Games Look Like Spinning Tops

**Wojciech Marian Czarnecki**
DeepMind
London

**Gauthier Gidel**
DeepMind
London

**Brendan Tracey**
DeepMind
London

**Karl Tuyls**
DeepMind
Paris

**Shayegan Omidshafiei**
DeepMind
Paris

**David Balduzzi**
DeepMind
London

**Max Jaderberg**
DeepMind
London

## Abstract

This paper investigates the geometrical properties of real world games (e.g. Tic-Tac-Toe, Go, StarCraft II). We hypothesise that their geometrical structure resembles a spinning top, with the upright axis representing transitive strength, and the radial axis representing the non-transitive dimension, which corresponds to the number of cycles that exist at a particular transitive strength. We prove the existence of this geometry for a wide class of real world games by exposing their temporal nature. Additionally, we show that this unique structure also has consequences for learning – it clarifies why populations of strategies are necessary for training of agents, and how population size relates to the structure of the game. Finally, we empirically validate these claims by using a selection of nine real world two-player zero-sum symmetric games, showing 1) the spinning top structure is revealed and can be easily reconstructed by using a new method of Nash clustering to measure the interaction between transitive and cyclical strategy behaviour, and 2) the effect that population size has on the convergence of learning in these games.

## 1 Introduction

Game theory has been used as a formal framework to describe and analyse many naturally emerging strategic interactions [30, 10, 9, 28, 20, 11, 6]. It is general enough to describe very complex interactions between agents, including classic real world games like Tic-Tac-Toe, Chess, Go, and modern computer-based games like Quake, DOTA and StarCraft II. Simultaneously, game theory formalisms apply to abstract games that are not necessarily interesting for humans to play, but were created for different purposes. In this paper we ask the following question: Is there a common structure underlying the games that humans find interesting and engaging?

Why is it important to understand the geometry of real world games? Games have been used as benchmarks for the development of artificial intelligence for decades, starting with Shannon's interest in Chess [27], through to the first reinforcement learning success in Backgammon [31], IBM DeepBlue [5] developed for Chess, and the more recent achievements of AlphaGo [29] mastering the game of Go, FTW [13] for Quake III: Capture the Flag, AlphaStar [34] for StarCraft II, OpenAI Five [23] for DOTA 2, and Pluribus [3] for no-limit Texas Hold 'Em Poker. We argue that grasping any common structures to these real world games is essential to understand why specific solution methods work, and can additionally provide us with tools to develop AI based on a deeper understanding of the scope and limits of solutions to previously tackled problems. The analysis of non-transitive behaviour has been critical for algorithm development in general game theoretic settings in the past [15, 1, 2]. Therefore a good tool to have would be the formalisation of non-transitive behaviour in real world games and a method of dealing with notion of transitive progress built on top of it.

We propose the Game of Skill hypothesis (Fig. 1) where strategies exhibit a geometry that resembles a spinning top, where the upright axis represents the transitive strength and the radial axis corresponds to cyclic, non-transitive dynamics. We focus on two aspects. Firstly, we theoretically and empirically validate whether the Games of Skill geometry materialises in real world games. Secondly, we unpack some of the key practical consequences of the hypothesis, in particular investigating the implications for training agents.

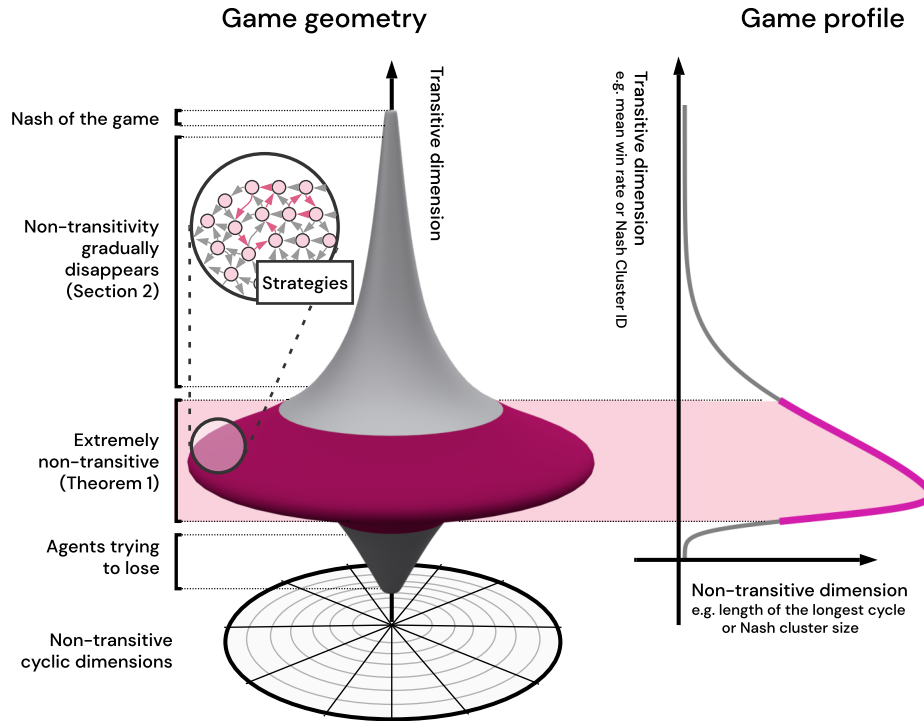

Figure 1: High-level visualisation of the geometry of Games of Skill. It shows a strong transitive dimension, that is accompanied by the highly cyclic dimensions, which gradually diminishes as skill grows towards the Nash Equilibrium (upward), and diminishes as skill evolves towards the worst possible strategies (downward). The simplest example of non-transitive behaviour is a cycle of length 3 that one finds e.g. in the Rock Paper Scissors game.

Some of the above listed works use multi-agent training techniques that are not guaranteed to improve/converge in all games. In fact, there are conceptually simple, yet surprisingly difficult cyclic games that cannot be solved by these techniques [2]. This suggests that a class of real world games might form a strict subset of 2-player symmetric zero-sum games, which are often used as a formalism to analyse such games. The Game of Skill hypothesis provides such a class, and makes specific predictions about how strategies behave. One clear prediction is the existence of tremendously long cycles, which permeate throughout the space of relatively weak strategies in each such game. Theorem 1 proves the existence of long cycles in a rich class of real world games that includes all the examples above. Additionally, we perform an empirical analysis of nine real world games, and establish that the hypothesised Games of Skill geometry is indeed observed in each of them.

Finally, we analyse the implications of the Game of Skill hypothesis for learning. In many of the works tackling real world games [13, 34, 23] some form of population-based training [12, 15] is used, where a collection of agents is gathered and trained against. We establish theorems connecting population size and diversity with transitive improvement guarantees, underlining the importance of population-based training techniques used in many of the games-related research above, as well as the notion of diversity seeking behaviours. We also confirm these with simple learning experiments over empirical games coming from nine real world games.

In summary, our contributions are three-fold: i) we define a game class that models real world games, including those studied in recent AI breakthroughs (e.g. Go, StarCraft II, DOTA 2); ii) we show both theoretically and empirically that a spinning top geometry can be observed; iii) we provide theoretical arguments that elucidate why specific state-of-the-art algorithms lead to consistent improvements in such games, with an outlook on developing new population-based training methods. Proofs are provided in Supplementary Materials B, together with details on implementations of empirical experiments (E, G, H), additional data (F), and algorithms used (A, C, D, I, J).

## 2 Game of Skill hypothesis

We argue that real world games have two critical features that make them Games of Skill. The first feature is the notion of progress. Players that regularly practice need to have a sense that they will improve and start beating less experienced players. This is a very natural property to keep people engaged, as there is a notion of skill involved. From a game theory perspective, this translates to a strong transitive component of the underlying game structure.

A game of pure Rock Paper Scissors (RPS) does not follow this principle and humans essentially never play it in a standalone fashion as a means of measuring strategic skill (without at least knowing the identity of their opponent and having some sense of their opponent's previous strategies or biases).

The second feature is the availability of diverse game styles. A game is interesting if there are many qualitatively different strategies [7, 17, 37] with their own strengths and weaknesses, whilst on average performing on a similar level in the population. Examples include the various openings in Chess and Go, which work well against other specific openings, despite not providing a universal advantage against all opponents. It follows that players with approximately the same *transitive skill level*, can still have imbalanced win rates against specific individuals within the group, as their game styles will counter one another. This creates interesting dynamics, providing players, especially at lower levels of skill, direct information on where they can improve. Crucially, this richness gradually disappears as players get stronger, so at the highest level of play, the outcome relies mostly on skill and less on game style. From a game theory perspective, this translates to non-transitive components that rapidly decrease in magnitude relative to the transitive component as skill improves.

These two features combined would lead to a cone-like shape of the game geometry, with a wide, highly cyclic base, and a narrow top of highly skilled strategies. However, while players usually play the game to win, the strategy space includes many strategies whose goal is to lose. While there is often an asymmetry between seeking wins and losses (it is often easier to lose than it is to win), the overall geometry will be analogous - with very few strategies that lose against every other strategy, thus creating a peaky shape at the bottom of our hypothesised geometry. This leads to a *spinning top* (Figure 1) – a geometry, where, as we travel across the transitive dimension, the non-transitivity first rapidly increases, and then, after reaching a potentially very large quantity (more formally detailed later), quickly reduces as we approach the strongest strategies. We refer to games that exhibit such underlying geometry as *Games of Skill*.

## 3 Preliminaries

We first establish preliminaries related to game theory and assumptions made herein. We refer to the options, or actions, available to any player of the game as a *strategy*, in the game-theoretic sense. Moreover, we focus on finite normal-form games (i.e. wherein the outcomes of a game are represented as a payoff tensor), unless otherwise stated.

We use $\Pi$ to denote the set of all strategies in a given game, with $\pi_i \in \Pi$ denoting a single pure strategy. We further focus on symmetric, deterministic, zero sum games, where the payoff (outcome of a game) is denoted by $\mathbf{f}(\pi_i, \pi_j) = -\mathbf{f}(\pi_j, \pi_i) \in [-1, 1]$. We say that $\pi_i$ beats $\pi_j$ when $\mathbf{f}(\pi_i, \pi_j) > 0$, draws when $\mathbf{f}(\pi_i, \pi_j) = 0$ and loses otherwise. For games which are not fully symmetric (e.g. all turn based games) we symmetrise them by considering a game we play once as player 1 and once as player 2. Many games we mention have an underlying time-dependent structure (e.g. chess); thus, it might be more natural to think about them in the so-called extensive-form, wherein player decision-points are expressed in a temporal manner. To simplify our analysis, we conduct our analysis by casting all such games to the normal-form, though we still exploit some of the time-dependent characteristics. Consequently, when we refer to a specific game (e.g. Tic-Tac-Toe), we also analyse

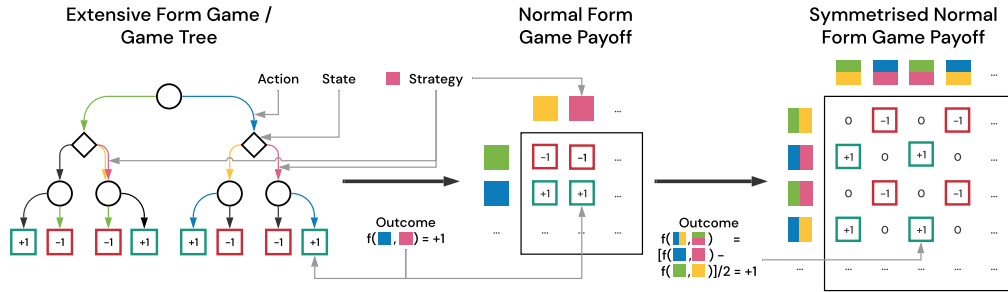

Figure 2: Left – extensive form/game tree representation of a simple 3-step game, where in each state a player can choose one of two actions, and after exactly 3 moves one of the players wins. Player 1 takes actions in circle nodes, and player 2 in diamond nodes. Outcomes are presented from the perspective of player 1. Middle – a partial normal form representation of this game, presenting outcomes for 4 strategies, colour coded on the graph representation. Right – a symmetrised version, where two colours denote which strategy one follows as player 1 and which as player 2.

the rules of the game itself, which might provide additional properties and insights into the geometry of the payoffs $\mathbf{f}$. In such situations, we explicitly mention that the property/insight comes from game *rules* rather than its payoff structure $\mathbf{f}$. This is somewhat different from a typical game theoretical analysis (for normal form games) that might equate game and $\mathbf{f}$. We use a standard tree representation of temporally extended games, where a node represents a state of the game (e.g. the board at any given time in the game of Tic-Tac-Toe), and edges represent what is the next game state when the player takes a specific action (e.g. spaces where a player can mark their $\times$ or $\circ$). The node is called terminal, when it is an end of the game and it provides an outcome $\mathbf{f}$. In this view a *strategy* is a deterministic mapping from states to actions, and an outcome between two strategies is simply the outcome of the terminal state they reach when they play against each other. Figure 2 visualises these views on an exemplary three step game.

We call a game *monotonic* when $\mathbf{f}(\pi_i, \pi_j) > 0$ and $\mathbf{f}(\pi_j, \pi_k) > 0$ implies $\mathbf{f}(\pi_i, \pi_k) > 0$. In other words, the relation of one strategy beating another is *transitive* in the set theory sense. We say that a set of strategies $\{\pi_i\}_{i=1}^l$ forms a cycle of length $l$ when for each $i > 1$ we have $\mathbf{f}(\pi_{i+1}, \pi_i) > 0$ and $\mathbf{f}(\pi_1, \pi_l) > 0$. For example, in the game of Rock Paper Scissors we have $\mathbf{f}(\pi_\mathrm{r}, \pi_\mathrm{s}) = \mathbf{f}(\pi_\mathrm{s}, \pi_\mathrm{p}) = \mathbf{f}(\pi_\mathrm{p}, \pi_\mathrm{r}) = 1$. There are various ways in which one could define a decomposition of a given game into the *transitive* and *non-transitive* components [2]. In this paper, we introduce Nash clustering, where the transitive component becomes an index of it, and non-transitivity corresponds to the size of this cluster. We do not claim that this is the only nor the best way of thinking about this phenomena, but we found it to have valuable mathematical properties.

The manner in which we study the geometry of games in this paper is motivated by the structural properties that AI practitioners have exploited to build competent agents for real world games [34, 29, 23], using reinforcement learning (RL). Specifically, consider an *empirical game-theoretic* outlook on training of policies in a game (e.g. Tic-Tac-Toe), where each trained policy (e.g. neural network) for a player is considered as a strategy of the empirical game. In other words, an empirical game is a normal-form game wherein AI policies are synonymous with strategies. Each of these policies, when deployed on the true underlying game, yields an outcome (e.g. win/loss) captured by the payoff in the empirical game. Thus, in each step of training, the underlying RL algorithm produces an approximate best response in the actual underlying (multistep, extensive form) game; this approximate best response is then added to the set of policies (strategies) in the empirical game, iteratively expanding it.

This AI training process is also often hierarchical – there is some form of multi-agent scheduling process that selects a set of agents to be beaten at a given iteration (e.g. playing against a previous version of an agent in self-play [29], or against some distribution of agents generated in the past [34]), and the underlying RL algorithm used for training new policies performs optimisation to find an agent that satisfies this constraint. There is a risk that the RL algorithm finds very weak strategies that

satisfy the constraint (e.g. strategies that are highly exploitable). Issues like this have been observed in various large-scale projects (e.g. exploits that human players found in the Open AI Five [23] or exploiters in League Training of AlphaStar [34]). This exemplifies some of the challenges of creating AI agents, which are not the same that humans face when they play a specific game. Given these insights, we argue that algorithms can be disproportionately affected by the existence of various non-transitive geometries, in contrast to humans.

# 4   Real world games are complex

The spinning top hypothesis implies that at some relatively low level of transitive strength, one should expect very long cycles in any Game of Skill. We now prove that, in a large class of games (ranging from board games such as Go and Chess to modern computer games such as DOTA and StarCraft), one can find tremendously long cycles, as well as any other non-transitive geometries.

We first introduce the notion of *n-bit communicative* games, which provide a mechanism for lower bounding the number of cyclic strategies. For a given game with payoff $\mathbf{f}$, we define its win-draw-loss version with the same rules and payoffs $\mathbf{f}^\dagger = \text{sign} \circ \mathbf{f}$, which simply removes the score value, and collapses all wins, draws, and losses onto +1, 0, and -1 respectively. Importantly, this transformation does not affect winning, nor the notion of cycles (though could, for example, change Nash equilibria).

**Definition 1.** *Consider the extensive form view of the win-draw-loss version of any underlying game; the underlying game is called n-bit communicative if each player can transmit $n \in \mathbb{R}_+$ bits of information to the other player before reaching the node whereafter at least one of the outcomes 'win' or 'loss' is not attainable.*

For example, the game in Figure 2 is 1-bit communicative, as each player can take one out of two actions before their actions would predetermine the outcome. We next show that as games become more communicative, the set of strategies that form non-transitive interactions grows exponentially.

**Theorem 1.** *For every game that is at least n-bit communicative, and every antisymmetric win-loss payoff matrix $\mathbf{P} \in \{-1,0,1\}^{\lfloor 2^n \rfloor \times \lfloor 2^n \rfloor}$, there exists a set of $\lfloor 2^n \rfloor$ pure strategies $\{\pi_1, ..., \pi_{\lfloor 2^n \rfloor}\} \subset \Pi$ such that $\mathbf{P}_{ij} = \mathbf{f}^\dagger(\pi_i, \pi_j)$, and $\lfloor x \rfloor = \max_{a \in \mathbb{N}} a \leq x$.*

In particular, this means that if we pick $\mathbf{P}$ to be cyclic – where for each $i < \lfloor 2^n \rfloor$ we have $\mathbf{P}_{ij} = 1$ for $j < i$, $\mathbf{P}_{ji} = -1$ and $\mathbf{P}_{ii} = 0$, and for the last strategy we do the same, apart from making it lose to strategy 1, by putting $\mathbf{P}_{\lfloor 2^n \rfloor 1} = -1$ – we obtain a constructive proof of a cycle of length $\lfloor 2^n \rfloor$, since $\pi_1$ beats $\pi_{\lfloor 2^n \rfloor}$, $\pi_{\lfloor 2^n \rfloor}$ beats $\pi_{\lfloor 2^n \rfloor - 1}$, $\pi_{\lfloor 2^n \rfloor - 1}$ beats $\pi_{\lfloor 2^n \rfloor - 2}$, ..., $\pi_2$ beats $\pi_1$. In practise, the longest cycles can be much longer (see the example of the Parity Game of Skill in the Supplementary Materials) and thus the above result should be treated as a lower bound.

Note, that strategies composing these long cycles will be very weak in terms of their transitive performance, but of course not as weak as strategies that actively seek to loose, and thus in the hypothesised geometry they would occupy the thick, middle level of the spinning top. Since such strategies do not particularly target winning or losing, they are unlikely to be executed by a human playing a game. Despite this, we use them to exemplify the most extreme part of the underlying geometry, and given that in both the extremes of very strong and very weak policies we expect non-transitivities to be much smaller than that, we hypothesise that they behave approximately monotonically in both these directions.

We provide an efficient algorithm to compute $n$ by traversing the game tree (linear in number of transitions between states) in Supplementary Materials together with derivation of its recursive formulation. We found that Tic-Tac-Toe is 5.58-bit communicative (which means that every payoff of size $47 \times 47$ is realised by some strategies). Additionally, all 1-step games (e.g. RPS) are 0-bit communicative, as all actions immediately prescribe the outcome without the ability to communicate any information. For games where state space is too large to be traversed, we can consider a heuristic choice of a subset of actions allowed in each state thus providing a lower bound on $n$, e.g. in Go we can play stones on one half of the board, and show that $n \geq 1000$.

**Proposition 1.** *The game of Go is at least 1000-bit communicative and contains a cycle of length at least $2^{1000}$.*

**Proposition 2.** *Modern games, such as StarCraft, DOTA or Quake, when limited to 10 minutes play, are at least 36000-bit communicative.*

The above analysis shows that real world games have an extraordinarily complex structure, which is not commonly analysed in classical game theory. The sequential, multistep aspect of these games makes a substantial difference, as even though one could simply view each of them in a normal form way [21], this would hide the true structure exposed via our analysis.

Naturally, the above does not prove that real world games follow the Games of Skill geometry. To validate the merit of this hypothesis, however, we simply follow the well-established path of proving hypothetical models in natural sciences (e.g. physics). Notably, the rich non-transitive structure (located somewhere in the middle of the transitive dimension) exposed by this analysis is a key property that the hypothesised Game of Skill geometry would imply. More concretely, in Section 6 we conduct empirical game theory-based analysis [33] of a wide range of real world games to show that the hypothesised spinning top geometry can, indeed, be observed.

## 5 Layered game geometry

The practical consequences of huge sets of non-transitive strategies are two-fold. First, building naive multi-agent training regimes, that try to deal with non-transitivity by asking agents to form a cycle (e.g. by losing to some opponents), is likely to fail – there are just too many ways in which one can lose without providing any transitive improvement for other agents trained against it. Second, there exists a shared geometry and structure across many games, that we should exploit when designing multi-agent training algorithms. In particular, we show how these properties justify some of the recent training techniques involving population-level play and the League Training used in Vinyals et al. [34]. In this section, we investigate the implications of such a game geometry on the training of agents, starting with a simplified variant that enables building of intuitions and algorithmic insights.

**Definition 2** $k$**-layered finite Game of Skill.** *We say that a game is a $k$-layered finite Game of Skill if the set of strategies $\Pi$ can be factorised into $k$ layers $L_i$ such that $\bigcup_i L_i = \Pi$, $\forall_{i \neq j} L_i \cap L_j = \emptyset$ and layers are fully transitive in the sense that $\forall_{i<j, \pi_i \in L_i, \pi_j \in L_j} \mathbf{f}(\pi_i, \pi_j) > 0$ and there exists $z \in \mathbb{N}$ such that for each $i < z$ we have $|L_i| \leq |L_{i+1}|$ and $|L_i| \geq |L_{i+1}|$ for $i \geq z$.*

Intuitively, all the non-transitive interaction take place within each layer $L_i$, whilst the skill (or transitive) component of the game corresponds to a layer ID. For every finite game, there exists $k \geq 1$ for which it is a $k$-layered game (though when $k = 1$ this structure is not useful). Moreover, every monotonic game has as many layers as there are strategies in the game. Even the simplest non-transitive structure can be challenging for many training algorithms used in practise [23, 29, 13], such as naive self-play [2]. However, a simple form of fictitious play with a hard limit on population size will converge independently of the oracle used (the oracle being the underlying algorithm that returns a new policy that satisfies a given improvement criterion):

**Proposition 3.** *Fixed-memory size fictitious play initialised with population of strategies $\mathcal{P}^0 \subset \Pi$ where at iteration $t$ one replaces some strategy in $\mathcal{P}^{t-1}$ with a new strategy $\pi$ such that $\forall_{\pi_i \in \mathcal{P}^{t-1}} \mathbf{f}(\pi, \pi_i) > 0$ converges in layered Games of Skill, if the population is not smaller than the size of the lowest layer occupied by at least one strategy in the population $|\mathcal{P}^0| \geq |L_{\arg\min_k : \mathcal{P}^0 \cap L_k \neq \emptyset}|$ and at least one strategy is above $z$. If all strategies are below $z$, then required size is that of $|L_z|$.*

Intuitively, to guarantee transitive improvements over time, it is important to cover all possible game styles. This proposition also leads to a known result of needing just one strategy in the population (e.g. self-play) to keep improving in monotonic games [2]. Finally, it also shows an important intuition related to how modern AI systems are built – the complexity of the non-transitivity discovery/handling methodology decreases as the overall transitive strength of the population grows. Various agent priors (e.g. search, architectural choices for parametric models such as neural networks, smart initialisation such as imitation learning etc.) will initialise in higher parts of the spinning top, and also restrict the set of representable strategies to the transitively stronger ones. This means that there exists a form of balance between priors one builds into an AI system and the amount of required multi-agent learning complexity required (see Figure 3 for a comparison of various recent state of the art AI systems). From a practical perspective, there is no simple way of knowing $|L_z|$ without traversing the entire game tree. Consequently, this property is not directly transferable to the design of an efficient algorithm (as if one had access to the full game tree traversal, one could simply use Min-Max to solve the game). Instead, this analysis provides an intuitive mechanism, explaining why finite-memory fictitious self-play can work well in practice.

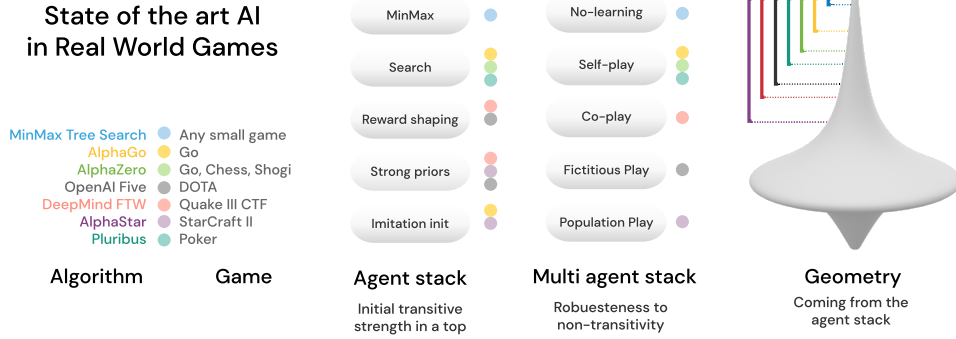

Figure 3: Visualisation of various state of the art approaches for solving real world games, with respect to the multi-agent algorithm and agent modules used (on the left). Under the assumption that these projects led to the approximately best agents possible, and that the Game of Skill hypothesis is true for these games, we can predict what part of the spinning top each of them had to explore (represented as intervals on the right). This comes from the complexity of the multi-agent algorithm (the method of dealing with non-transitivity) that was employed – the more complex the algorithm, the larger the region of the top that was likely represented by the strategies using the specific agent stack. This analysis does not expose which approach is better or worse. Instead, it provides intuition into how the development of training pipelines used in the literature enables simplification of non-transitivity avoidance techniques, as it provides an initial set of strategies high enough in the spinning top.

In practise, the non-transitive interactions are not ordered in a simple layer structure, where each strategy from one beats each from the other. We can however relax notion of transitive relation which will induce a new cluster structure. The idea behind this approach, called *Nash clustering*, is to first find the mixed Nash equilibrium of the game payoff $\mathbf{P}$ over the set of pure strategies $\Pi$ (we denote the equilibrium for payoff $\mathbf{P}$ when restricted only to strategies in $X$ by $\mathrm{Nash}(\mathbf{P}|X)$), and form a first cluster by taking all the pure strategies in the support of this mixture. Then, we restrict our game to the remaining strategies, repeating the process until no strategies remain.

**Definition 3.** *Nash clustering* $\mathrm{C}$ *of the finite zero-sum symmetric game strategy* $\Pi$ *set by setting for each* $i \geq 1$: $N_{i+1} = \mathrm{supp}(\mathrm{Nash}(\mathbf{P}|\Pi \setminus \bigcup_{j \leq i} N_j))$ *for* $N_0 = \emptyset$ *and* $\mathrm{C} = (N_j : j \in \mathbb{N} \wedge N_j \neq \emptyset)$.

While there might be many Nash clusterings per game, there exists a unique maximum entropy Nash clustering where at each iteration we select a Nash equilibrium with maximum Shannon entropy, which is guaranteed to be unique [24] due to the convexity of the objective. The crucial result is that Nash clusters form a monotonic ordering with respect to Relative Population Performance (RPP) [2], which is defined for two sets of agents $\Pi_A, \Pi_B$ with a corresponding Nash equilibrium of the asymmetric game $(p_A, p_B) := \mathrm{Nash}(\mathbf{P}_{AB}|(A, B))$ as $\mathrm{RPP}(\Pi_A, \Pi_B) = p_A^\mathrm{T} \cdot \mathbf{P}_{AB} \cdot p_B$.

**Theorem 2.** *Nash clustering satisfies* $\mathrm{RPP}(\mathrm{C}_i, \mathrm{C}_j) \geq 0$ *for each* $j > i$.

We refer to this notion as a relaxation, since it is not each strategy in one cluster that is better than in the other, but rather the whole cluster is better than the other. In particular, this means that in $k$-layered game, the new clusters are subsets of layers (because Nash equilibrium will never contain a fully dominated strategy). Next we show that a diverse population that spans an entire cluster guarantees transitive improvement, despite not having access to any weaker policies nor knowledge of covering the cluster.

**Theorem 3.** *If at any point in time, the training population* $\mathcal{P}^t$ *includes any full Nash cluster* $\mathrm{C}_i \subset \mathcal{P}^t$, *then training against* $\mathcal{P}^t$ *by finding* $\pi$ *such that* $\forall_{\pi_j \in \mathcal{P}^t} \mathbf{f}(\pi, \pi_j) > 0$ *guarantees transitive improvement in terms of the Nash clustering* $\exists_{k<i} \ \pi \in \mathrm{C}_k$.

Consequently, to keep improving transitively, it is helpful to seek wide coverage of strategies around the current transitive strength (inside the cluster). This high level idea has been applied in some multi-player games such as soccer [18] and more recently StarCraft II. AlphaStar [34] explicitly

attempts to cover the non-transitivities using exploiters, which implicitly try to expand on the current Nash. Interestingly, same principle can be applied to single-player domains and justify seeking diversity of the environments, so that agents need to improve transitively with respect to them. With the Game of Skill geometry one can rely on this required coverage to be smaller over time (as agents get stronger). Thus, forcing the new generation of agents to be the weakest ones that beat the previous one would be sufficient to keep covering cluster after cluster, until reaching the final one.

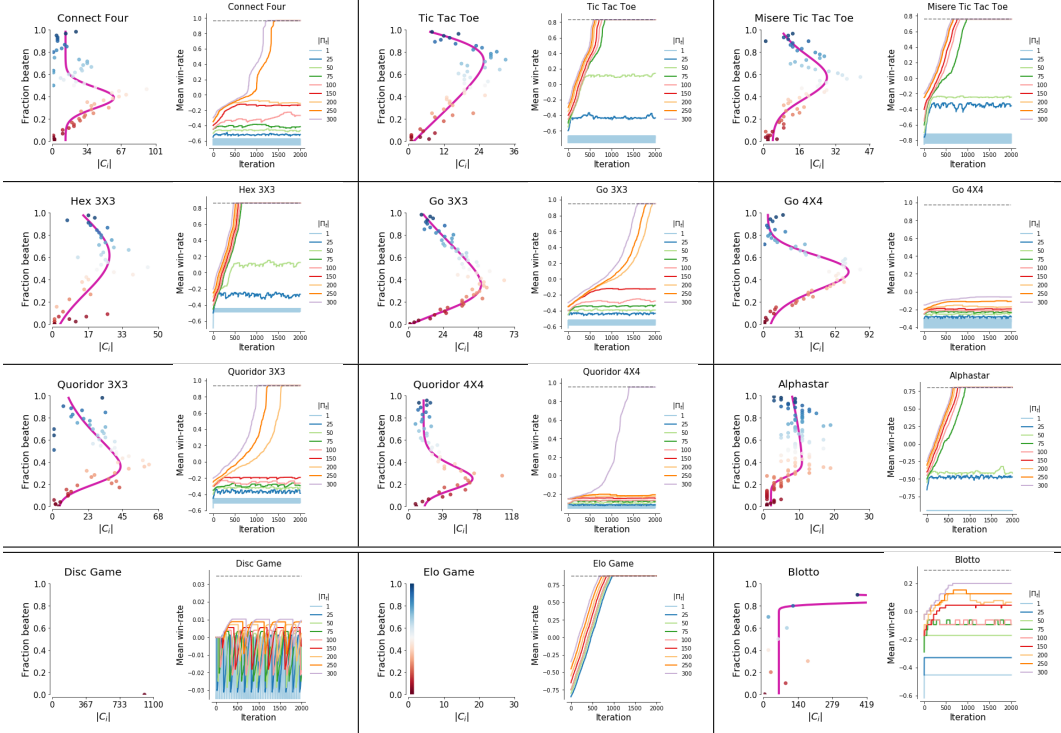

Table 1: (Left of each plot) Game profiles of empirical game geometries, when sampling strategies in various real world games, such as Connect Four, Tic-Tac-Toe and StarCraft II (note that strategies in AlphaStar come from a learning system, and not our sampling strategy, see Supplementary Materials for details and discussion). The first three rows shows clearly the Game of Skill geometry, while the last row shows the geometry for games that are not Games of Skill, and clearly do not follow this geometry. The pink curve shows a fitted Skewed Gaussian highlighting the spinning top shape (details in Supplementary Materials). (Right of each plot) Learning curves in empirical games, using various population sizes, the oldest strategy in the population is replaced with one that beats the whole population on average using an adversarial oracle (returning the weakest strategy satisfying this goal). For Games of Skill there is a phase change of behaviour for most games, where once the population is big enough to deal with the non transitivity, the system converges to the strongest policy. On the other hand, in other games (bottom) such as the Disc game, no population size avoids cycling, and for fully transitive games like the Elo game, even naive self play converges.

## 6 Empirical validation of Game of Skill hypothesis

To empirically validate the spinning top geometry, we consider a selection of two-player zero-sum games available in the OpenSpiel library [16]. Unfortunately, even for the simplest of real world games, the strategy space can be enormous. For example, the number of behaviourally unique pure strategies in Tic-Tac-Toe is larger than $10^{567}$ (see Supplementary Materials). A full enumeration-based analysis is therefore computationally infeasible. Instead, we rely on empirical game-theoretic analysis, an experimental paradigm that relies on simulation and sampling of strategies to construct abstracted counterparts of complex underlying games, which are more amenable for analysis [35, 36, 25, 38, 26, 32]. Specifically, we look for strategy sampling that covers the strategy space as uniformly as possible so that the underlying geometry of the game (as exposed by the

empirical counterpart) is minimally biased. A simple and intuitive procedure for strategy sampling is as follows. First, apply a tree-search method, in the form of Alpha-Beta [22] and MCTS [4] and select a range of parameters that control the transitive strength of these algorithms (depth of search for Alpha-Beta and number of simulations for MCTS) to ensure coverage of transitive dimension. Second, for each such strategy we create multiple instances, with varied random number seed, thus causing them to behave differently. We additionally include Alpha-Beta agents that actively seek to lose, to ensure discovery of the lower cone of the hypothesised spinning top geometry. While this procedure does not guarantee uniform sampling of strategies, it at least provides decent coverage of the transitive dimension. In total, this yields approximately 1000 agents per game. Finally, following strategy sampling, we form an empirical payoff table with entries evaluating the payoffs of all strategy match-ups, remove all duplicate agents, and use this matrix to approximate the underlying game of interest.

Table 1 summarises the empirical analysis which, for the sake of completeness, includes both Games of Skill and games that are *not* Games of Skill such as the Disc game [2], a purely transitive Elo game, and the Blotto game. Overall, all real world games results show the hypothesised spinning top geometry. More closely inspecting the example of Go ($3\times3$) in Table 2 of the Supplementary Materials, we notice that the Nash clusters induced payoff look monotonic, and the sizes of these are maximal around the mid-ranges of transitive strength, and quickly decrease as transitive strength both increases or decreases. At the level of the strongest strategies, non-trivial Nash clusters exist, showing that even in this empirical approximation of the game of Go on a small board, one still needs some diversity of play styles. This is to be expected due to various game symmetries of the game rules. Moreover, various games that were created to study game theory (rather than for humans to play) fail to exhibit the hypothesised geometry. In the game of Blotto, for example, the size of Nash clusters keep increasing, as the number of strategies one needs to mix at higher and higher levels of play in this game keeps growing. This is a desired property for the purpose of studying complexity of games, but arguably not so for a game that is simply played for enjoyment. In particular, the game of Blotto requires players to mix uniformly over all possible permutations to be unexploitable (since the game is invariant to permutations), which is difficult for a human player to achieve.

We tested the population size claims of Nash coverage as follows. First, construct empirical games coming from the sampling of $n$ agents defined above, yielding an approximation of the underlying games. Second, define a simple learning algorithm, where we start with $k$ (size of the population) weakest strategies (wrt. mean win-rate) and iteratively replace the oldest one with a strategy $\pi$ that beats the entire population $\mathcal{P}^t$ on average, meaning that $\sum_{\pi'\in\mathcal{P}^t} \mathbf{f}(\pi,\pi') > 0$. To pick the new strategy, we use a pessimistic oracle that selects the weakest strategy satisfying the win-rate condition. This counters the bias towards sampling stronger strategies, thus yielding a more fair approximation of typical greedy learning methods such as gradient-based methods or reinforcement learning.

For small population sizes, training does not converge and cycles for all games (Table 1). As the population grows, strength increases but saturates in various suboptimal cycles. However, when the population exceeds a critical size, training converges to the best strategies in almost all experiments. For games that are not real world games we observe quite different behaviour - where, despite growth of population size, cycling keeps occuring (e.g. the Disc game), convergence is guaranteed even with a population of size 1 (e.g. the Elo game, which is monotonic).

## 7   Conclusions

In this paper we have introduced Games of Skill, a class of games that, as motivated both theoretically and empirically, includes many real world games, including Tic-Tac-Toe, Chess, Go and even StarCraft II and DOTA. In particular we showed, that $n$-step games have tremendously long cycles, and provided both mathematical and algorithmic methods to estimate this quantity. We showed, that Games of Skill have a geometry resembling a spinning top, which can be used to reason about their learning dynamics. In particular, our insights provide useful guidance for research into population-based learning techniques building on League training [34] and PBT [13], especially when enriched with notions of diversity seeking [2]. Interestingly, we show that many games from classical game theory are *not* Games of Skill, and as such might provide challenges that are not necessarily relevant to developing AI methods for real world games. We hope that this work will encourage researchers to study real world games structures, to build better AI techniques that can exploit their unique geometries.

## Broader Impact

This work focuses on better understanding of mathematical properties of real world games and how they could be used to understand successful AI techniques that were developed in the past. Since we focus on retrospective analysis of a mathematical phenomenon, on exposing an existing structure, and deepening our understanding of the world, we do not see any direct risks it entails. Introduced notions and insights could be used to build better, more engaging AI agents for people to play with in real world games (e.g. AIs that grow with the player, matching their strengths and weaknesses). In a broader spectrum, some of the insights could be used for designing and implementing new games, that humans would fine enjoyable though challenges they pose. In particular it could be a viewed as a model for measuring how much notion of progress the game consists of. However, we acknowledge that methods enabling improved analysis of games may be used for designing products with potentially negative consequences (e.g., games that are highly addictive) rather than positive (e.g., games that are enjoyable and mentally developing).

## Acknowledgements

We would like to thank Alex Cloud for valuable comments on empirical analysis of the AlphaStar experiment, which lead to adding an extended section in the Supplementary Materials. We are also thankful to authors of OpenSpiel framework for the help provided with setting up the experiments that allowed empirical validation of the hypothesised geometry. The authors would also like to thank Gema Parreño Piqueras for insights into presentation and visualisations of the paper that allowed us to improve figures as well as the way concepts are presented.

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
