[Supplementary Material]

# Real World Games Look Like Spinning Tops Supplementary Materials

## A    Proofs

**Theorem 1.** *For every game that is at least $n$-bit communicative, and every antisymmetric win-loss payoff matrix $\mathbf{P} \in \{-1, 0, 1\}^{\lfloor 2^n \rfloor \times \lfloor 2^n \rfloor}$, there exists a set of $\lfloor 2^n \rfloor$ pure strategies $\{\pi_1, ..., \pi_{\lfloor 2^n \rfloor}\} \subset \Pi$ such that $\mathbf{P}_{ij} = \mathbf{f}^\dagger(\pi_i, \pi_j)$, and $\lfloor x \rfloor = \max_{a \in \mathbb{N}} a \leq x$.*

*Proof.* Let us assume we are given some $\mathbf{P}_{ij}$. We define corresponding strategies $\pi_i$ such that each starts by transmitting its ID as a binary vector using **n** bits. Afterwards, strategy $\pi_i$ reads out $\mathbf{P}_{ij}$ based on its own id, as well as the decoded ID of an opponent $\pi_j$, and since we assumed each win-draw-loss outcome can still be reached in a game tree, players then play to win/draw or lose, depending on the value of $\mathbf{P}_{ij}$. We choose $\pi_i$ and $\pi_j$ to follow the first strategy in lexicographic ordering (to deal with partially observable/concurrent move games) over sequences of actions that leads to $\mathbf{P}_{ij}$ to guarantee the outcome. Ordering over actions is arbitrary and fixed. Since identities are transmitted using binary codes, there are $\lfloor 2^n \rfloor$ possible ones. $\square$

**Proposition 1.** *The game of Go is at least 1000-bit communicative and contains a cycle of length at least $2^{1000}$.*

*Proof.* Since Go has a *resign* action, one can use the entire state space for information encoding, whilst still being able to reach both winning and losing outcomes. The game is played on a $19 \times 19$ board – if we split it in half we get 180 places to put stones per side, such that the middle point is still empty, and thus any placement of players stones on their half is legal and no stones die. These 180 fields give each player the ability to transfer $\sum_{i=180}^{1} \log_2(i) = \log_2(180!) \approx 1000$ bits. and according to Theorem 1 we thus have a cycle of length $2^{1000} > 10^{100}$. Figure 6 provides visualisation of this construction. $\square$

**Proposition 2.** *Modern games, such as StarCraft, DOTA or Quake, when limited to 10 minutes play, are at least 36000-bit communicative.*

*Proof.* With modern games running at 60Hz, as long as agents can "meet" in some place, and execute 60 actions per second that does not change their visibility (such as tiny rotations), they can transmit $60 \cdot 60 \cdot 10 = 36000$ bits of information per 10 minute encounter. Note, that this is very loose lower bound, as we are only transmitting one bit of information per action, while this could be significantly enriched, if we allow for use of multiple actions (such as jumping, moving multiple units etc.). $\square$

**Proposition 3.** *Fixed-memory size fictitious play initialised with population of strategies $\mathcal{P}^0 \subset \Pi$ where at iteration $t$ one replaces some strategy in $\mathcal{P}^{t-1}$ with a new strategy $\pi$ such that $\forall_{\pi_i \in \mathcal{P}^{t-1}} \mathbf{f}(\pi, \pi_i) > 0$ converges in layered Games of Skill, if the population is not smaller than the size of the lowest layer occupied by at least one strategy in the population $|\mathcal{P}^0| \geq |\mathrm{L}_{\arg\min_k : \mathcal{P}^0 \cap \mathrm{L}_k \neq \emptyset}|$ and at least one strategy is above $z$. If all strategies are below $z$, then required size is that of $|\mathrm{L}_z|$.*

*Proof.* Let's assume at least one strategy is above $z$. We will prove, that there will be at most $|\mathcal{P}^t| - 1$ consecutive iterations where algorithm will not improve transitively (defined as a new strategy being part of $\mathrm{L}_i$ where $i$ is smaller than the lowest number of all $\mathrm{L}_j$ that have non empty intersections with $\mathcal{P}^t$). Since we require the new strategy $\pi_{t+1}$ added at time $t+1$ to beat all previous strategies, it has to occupy at least a level, that is occupied by the strongest strategy in $\mathcal{P}^t$. Let's denote this level by $\mathrm{L}_k$, then $\pi_{t+1}$ improves transitively, meaning that there exists $i < k$ such that $\pi_{t+1} \in \mathrm{L}_i$, or it belongs to $\mathrm{L}_k$ itself. Since by construction $|\mathrm{L}_k| \leq |\mathcal{P}^t|$, this can happen at most $|\mathcal{P}^t| - 1$ times, as each strategy in $\mathcal{P}^t \cap \mathrm{L}_k$ needs to be beaten by $\pi_{t+1}$ and $|\mathcal{P}^t \cap \mathrm{L}_k| < |\mathcal{P}^t|$. By the analogous argument, if all the strategies are below $\mathrm{L}_z$, one can have at most $|\max_i |\mathrm{L}_i| - 1$ consecutive iterations without transitive improvement. $\square$

**Theorem 2.** *Nash clustering satisfies $\mathrm{RPP}(\mathrm{C}_i, \mathrm{C}_j) \geq 0$ for each $j > i$.*

*Proof.* By definition for each $A$ and each $B' \subset B$ we have $\text{RPP}(A, B') \geq \text{RPP}(A, B)$, thus for $X_i := \Pi \setminus \bigcup_{k<i} C_k$ and every $j > i$ we have $C_j \subset X_i$ and

$$
\begin{aligned}
\text{RPP}(C_i, C_j) &\geq \text{RPP}(C_i, X_i) \\
&= \text{RPP}(\text{supp}(\text{Nash}(\mathbf{P}|X_i)), X_i) \\
&= \text{RPP}(X_i, X_i) = 0.
\end{aligned} \tag{1}
$$

$\square$

**Theorem 3.** *If at any point in time, the training population $\mathcal{P}^t$ includes any full Nash cluster $C_i \subset \mathcal{P}^t$, then training against $\mathcal{P}^t$ by finding $\pi$ such that $\forall_{\pi_j \in \mathcal{P}^t} \mathbf{f}(\pi, \pi_j) > 0$ guarantees transitive improvement in terms of the Nash clustering $\exists_{k<i} \pi \in C_k$.*

*Proof.* Lets assume that $\exists_{k>i} \pi \in C_k$. This means, that

$$
\text{RPP}(C_i, C_k) \leq \max_{\pi_j \in C_i} \mathbf{f}(\pi_j, \pi) = \max_{\pi_j \in C_i}[-\mathbf{f}(\pi, \pi_j)] < 0, \tag{2}
$$

where the last inequality comes from the fact that $C_i \subset \mathcal{P}^t$ and $\forall_{\pi_j \in \mathcal{P}^t} \mathbf{f}(\pi, \pi_j) > 0$ implies that $\forall_{\pi_j \in C_i} \mathbf{f}(\pi, \pi_j) > 0$. This leads to a contradiction with the Nash clustering and thus $\pi \in C_k$ for some $k \leq i$. Finally $\pi$ cannot belong to $C_i$ itself since $\forall_{\pi_j \in C_i} \mathbf{f}(\pi, \pi_j) > 0 = \mathbf{f}(\pi, \pi)$. $\square$

# B   Computing $n$ in $n$-bit communicative games

Our goal is to be able to encode identity of a pure strategy in actions it is taking, in such a way, that opponent will be able to decode it. We focus on fully observable, turn-based games. Note, that with pure policy, and fully observable game, the only way to sent information to the other player is by taking an action (which is observed). Consequently, if at given state one considers $A$ actions, then choosing one of them we can transmit $\log_2(A)$ bits. We will build our argument recursively, by considering subtrees of a game tree. Naturally, a subtree is a tree of some game. Since the assumption of $n$-bit communicativeness is that we can transmit $n$ bits of information before outcomes become independent, it is easy to note that a subtree for which we cannot find terminal nodes with both outcomes (-1, +1) is 0-bit communicative. Let's remove these nodes from the tree. In the new tree, all the leaf nodes are still 0-bit communicative, as now they are "one action away" from making the outcome deterministic. Let's define function $\phi$ per state, that will output how many bits each player can transmit, before the game becomes deterministic, so for each player $j$

$$
\phi_j(s) = 0 \text{ if } s \text{ is a leaf.}
$$

The crucial element is how to now deal with a decision node. Let's use notation $c(s)$ to denote set of all children states, which we assume correspond to taking actions available in this state. If many actions would lead to the same state, we just pretend only one such action exists. From the perspective of player $j$, what we can do, is to select a subset of states that are reachable from $s$. If we do so, we will be able to encode $\log_2 |c(s)|$ bits in this move plus whatever we can encode in the future, which is simply $\min_{s' \in c(s)} \phi_j(s')$ as we need to guarantee being able to transmit this number of bits no matter which path is taken.

$$
\phi_j(s) = \max_{I \subset c(s)} \left\{ \log_2 |c(s)| + \min_{s' \in c(s)} \phi_j(s') \right\}
$$

However, our argument is symmetric, meaning that we need to not only transmit bits as player $j$, but also our opponent, and to do so we need to consider minimum over players respective communication channels:

$$
\phi_j(s) = \max_{I \subset c(s)} \left\{ \log_2 |c(s)| + \min_{s' \in c(s)} \min_i \phi_i(s') \right\}
$$

It is easy to notice that for a starting state $s_0$ we now have that the game is $\min_i \phi_i(s_0)$-bit communicative. The last recursive equation might look intractable, due to iteration over subsets of children states. However, we can easily compute quantities like this in linear time. Let's take general form of

$$
\max_{A \subset B} \{ g_0(|A|) + \min_{a \in A} g_1(a) \} =: \max_{A \subset B} g(A) \tag{3}
$$

Table 2: Game profiles of empirical game geometries, when sampling strategies in various real world games, such as Connect Four, Tic Tac Toe and even StarCraft II. The first three rows shows clearly the Game of Skill geometry, while the last row shows the geometry for games that are not Games of Skill, and clearly do not follow this geometry. Rows of the payoffs are sorted by mean winrate for easier visual inspection. The pink curve shows a fitted Skewed Gaussian to show the spinning top shape, details provided in Supplementary Materials.

and let's consider Alg. 1. To prove that it outputs maximum of $g$, let's assume that at any point

---

**Algorithm 1** Solver for Eq. 3 in $\mathcal{O}(|B|)$.

---

**Input:** functions $f$, $g$ and set $B$:
**begin**
$C \leftarrow \emptyset$
$g(X) \leftarrow g_0(|X|) + \min_{x \in X} g_1(x)$ {Eq. 3}
**sort** $B$ **in descending order of** $g_1$
**for** $b \in B$ **do**
  **if** $g(C \cup \{b\}) > g(C)$ **then**
    $C \leftarrow C \cup \{b\}$
  **end if**
**end for**
**return** $C$

---

$t$ we decided to pick $b' \neq b_t$. Since $b_t$ has highest $g$ at this point, we have $g_1(b') < g_1(b_t)$, and consequently $g(C_{t-1} \cup \{b'\}) < g(C_{t-1} \cup \{b_t\})$ so we decreased function value and conclude optimality proof.

We provide a pseudocode in Alg. 2 for the two-player, turn-based case with deterministic transitions. Analogous construction will work for $k$ players, simultaneous move games, as well as games with chance nodes (one just needs to define what we want to happen there, taking minimum will guarantee transmission of bits, and taking expectation will compute expected number of bits instead).

Exemplary execution at some state of Tic-Tac-Toe is provided in Figure 5. Figure 6 shows the construction from Proposition 1 for the game of Go.

We can use exactly the same procedure to compute $n$-communicativeness over restricted set of policies. For example let us consider strategies using MinMax algorithm to a fixed depth, between 0 and 9. Furthermore, we restrict what kind of first move they can make (e.g. only in the centre, or in a way that is rotationally invariant). Each such class simply defines a new "leaf" labelling of our tree or set of available children. Once we reach a state, after which considered policy is deterministic, by definition its communicativeness is zero, so we put $\phi(s) = 0$ there. Then we again run the recursive procedure. Running this analysis on the game of Tic-Tac-Toe (Fig. 4) reveals the Spinning Top like geometry wrt. class of policies used. As MinMax depth grows, cycle length bound from Theorem 1 decreases rapidly. Similarly introducing more inductive bias in the form of selecting what are good first moves affect the shape in an analogous way. This example has two important properties. First, it

Figure 4: Visualisation of cycle bound lengths coming from Theorem 1, when applied to the game of Tic-Tac-Toe over restricted set of policies – y axis corresponds to the depth of MinMax search (encoding transitive strength); and colour and line style correspond to restricted first move (encoding better and better inductive prior over how to play this game).

shows cyclic dimensions behaviour over whole policy space, as we do not rely on any sampling, but rather consider the whole space, restricting the transitive strength and using Theorem 1 as a proxy of non-transitivity. Second, it acts as an exemplification of the claim of various inductive biases restricting the part of the spinning top one needs to deal with when developing and AI for the specific game.

**Algorithm 2** Main algorithm to compute $n$ for which a given fully observable two-player zero-sum game is $n$-bit communicative.

---

**Input:** Game tree encoded with:
- states: $s_i \in \mathcal{S}$
- value of a state: $v(s_i) \in \{-1, 0, +1, \emptyset\}$
- set of children states $c(s_i) \subset \mathcal{S}$
- set of parent states $d(s_i) \subset \mathcal{S}$
- which player moves $p(s_i) \in \{0, 1\}$
**begin** {Remove states with deterministic outcomes}
$s_i \leftarrow \{s_i : \forall_{o \in \{-1, +1\}} \exists_{\text{path}(s_i, s_j)} \wedge v(s_j) = o\}$
**update** $c$
$q = [s_i : c(s_i) = \emptyset]$ {Init with leaves}
**while** $|q| > 0$ **do**
    $x \leftarrow q.\text{pop}()$
    $\phi(x) = \mathbf{Agg}(x)$ {Alg. 3}
    **for** $y \in d(x)$ **do**
        **if** $\forall z \in c(p)$ defined$(\phi(z))$ **then**
            $q.\text{enqueue}(y)$ {Enqueue a parent if all its children were analysed}
        **end if**
    **end for**
**end while**
**return** $\min \phi(s_0)$

---

**Algorithm 3** Aggregate (**Agg**) - helper function for Alg. 2

---

**Input:** State $x$
**begin**
$m \leftarrow [\min \phi(z) \text{ for } z \in c(x)]$ {min over players}
$o \leftarrow [\phi(z)[1 - p(x)] \text{ for } z \in c(x)]$ {other player bits}
**sort** $m$ **in decreasing order** {Order by decreasing communicativeness}
**order** $o$ **in the same order**
$b \leftarrow (0, 0)$
**for** $i = 1$ **to** $|c(x)|$ **do**
    $t[p(x)] \leftarrow \min(m[:i]) + \log_2(i)$
    $t[1 - p(x)] \leftarrow \min(o[:i])$
    **if** $t[p(x)] > b[p(x)]$ **then**
        $b = t$ {Update maximum}
    **end if**
**end for**
**return** $b$

---

# C   Cycles counting

In general even the problem of deciding if a graph has a simple path of length higher than some (large) $k$ is NP-hard. Consequently we focus our attention only on cycles of length 3 (which embed Rock-Paper-Scissor dynamics). For this problem, we can take adjacency matrix $A_{ij} = 1 \iff \mathbf{P}_{ij} > 0$ and simply compute diag$(A^3)$, which will give us number of length 3 cycles that pass through each node. Note, that this technique no longer works for longer cycles as diag$(A^p)$ computes number of closed walks instead of closed paths (in other words – nodes could be repeated). For $p = 3$ these concepts coincide though.

# D   Nash computation

We use iterative maximum entropy Nash solver for both Nash clustering and RPP [2] computation. Since we use numerical solvers, the mixtures found are not exactly Nash equilibria. To ensure that

they are "good enough" we find a best response, and check if the outcome is bigger than -1e-4. If it fails, we continue iterating until it is satisfied. For the data considered, this procedure always terminated. While usage of maximum entropy Nash might lead to unnecessarily "heavy" tops of the spinning top geometry (since equivalently we could pick smallest entropy ones, which would form more peaky tops) it guaranteed determinism of all the procedures (as maximum entropy Nash is unique).

# E   Games/payoffs definition

After construction of each empirical payoff $\mathbf{P}$, we first symmetrise it (so that ordering of players does not matter), and then standarise it $\mathbf{P'}_{ij} := \frac{\mathbf{P}_{ij} - \mathbf{P}_{ji}}{2 \max |\mathbf{P}|}$ for the analysis and plotting to keep all the scales easy to compare. This has no effect on Nashes or transitive strength, and is only used for consistent presentation of the results, as $\mathbf{P'} \in [-1, 1]^{N \times N}$. For most of the games this was an identity operation (as for most $\mathbf{P}$ we had $\max \mathbf{P} = -\min \mathbf{P} = 1$), and was mostly useful for various random games and Blotto, which have wider range of outcomes.

## E.1   Real world games

We use OpenSpiel [16] implementations of all the games analysed in this paper, with following setups:

- Hex 3X3: `hex(board_size=3)`
- Go 3X3: `go(board_size=3,komi=6.5)`
- Go 4X4: `go(board_size=4,komi=6.5)`
- Quoridor 3X3: `quoridor(board_size=3)`
- Quoridor 4X4: `quoridor(board_size=4)`
- Tic Tac Toe: `tic_tac_toe()`
- Misere Tic Tac Toe (a game of Tic Tac Toe where one wins if and onlfy if opponent makes a line): `misere(game=tic_tac_toe())`
- Connect Four: `connect_four()`

## E.2   StarCraft II (AlphaStar)

We use payoff matrix of the League of the AlphaStar Final [34] which represent a big population (900 agents) playing at a wide range of skills, using all 3 races of the game, and playing it without any simplifications. We did not run any of the StarCraft experiments. Sampling of these strategies is least controlled, and comes from a unique way in which AlphaStar system was trained.

This heavily skewed strategies sampling means that what we are observing is a study of AlphaStar induced game geometry, rather then necesarily geometry of the StarCraft II itself. In particular, one can ask why do we see a spinning top shape, rather than an upper cone, that we might expect given that AlphaStar agents never try to lose. The answer lies in how these strategies were created [34] namely – they come from iterative process, where agents are trained to beat all the previous strategies. In such setup, despite lack of an agent actively seeking to lose, the initial strategies will act as if they were designed to do so, since every other strategy was trained to beat them, while they were never trained to defend. The non-transitivies start to emerge, once "League exploiters" and "Exploiters" are slowly added to the population, and thus building strategic diversity. While these two factors and dynamics are different from the ones that motivate the geometry in remaining experiments, it surprisingly shared the self-similarity. From the perspective of the entire game of StarCraft II however, the shape we are observing is slightly warped, and we would expect to see an upper cone, if we were given ability to sample weak strategies more uniformly, without every other strategy being sampled conditionally on beating them.

### E.3  Rock Paper Scissor (RPS)

We use standard Rock-Paper-Scissor payoff of form

$$\mathbf{P} = \begin{bmatrix} 0 & 1 & -1 \\ -1 & 0 & 1 \\ 1 & -1 & 0 \end{bmatrix}.$$

This game is fully cyclic, and there is no pure strategy Nash (the only Nash-equilibrium is the uniform mixture of strategies).

Maybe surprisingly, people do play RPS competitively, however it is important to note, that in "real-life" the game of RPS is much richer, than its game theoretic counterpart. First, it often involves repeated trials, which means one starts to reason about the strategy opponent is employing, and try to exploit it while not being exploited themselves. Second, identity of the opponent is often known, and since player are humans, they have inherit biases in the form of not being able to play completely randomly, having beliefs, preferences and other properties, that can be analysed (based on historical matches) and exploited. Finally, since the game is often played in a physical environment, there might be various subconscious tells for a given player, that inform the opponent about which move they are going to play, akin to Clever Hans phenomena.

### E.4  Disc Game

We use definition of random game from the "Open-ended learning in symmetric zero-sum games" paper [2]. We first sample $N = 1000$ points uniformly in the unit circle $A_i \sim U(S(0, 1))$ and then put

$$\mathbf{P}_{ij} = A_i^{\mathrm{T}} \begin{bmatrix} 0 & -1 \\ 1 & 0 \end{bmatrix} A_j.$$

Similarly to RPS, this game is fully cyclic.

### E.5  Elo game

We sample Elo rating [8] per player $S_i \sim \mathcal{U}(0, 2000)$, and then put $\mathbf{P}_{ij} := (1 + e^{-(S_i - S_j)/400})^{-1}$, which is equivalent of using scaled difference in strength $D_{ij} = (S_i - S_j)/400$ squashed through a sigmoid function $\sigma(x) = (1 + e^{-x})^{-1}$. It is easy to see that this game is monotonic, meaning that $\mathbf{P}_{ij} > \mathbf{P}_{jk} \rightarrow \mathbf{P}_{ik}$. We use $N = 1000$ samples.

### E.6  Noisy Elo games

For a given noise $\epsilon > 0$ we first build an Elo game, and then take $N^2$ independent samples from $\mathcal{N}(0, \epsilon)$ and add it to corresponding entries of $\mathbf{P}$, creating $\mathbf{P}_\epsilon$. After that, we symmetrise the payoff by putting $\mathbf{P} := \mathbf{P}_\epsilon - \mathbf{P}_\epsilon^{\mathrm{T}}$.

### E.7  Random Game of Skill

We put $\mathbf{P}_{ij} := \frac{1}{2}(W_{ij} - W_{ji}) + S_i - S_j$ where each of the random variables $W_{ij}, S_i$ comes from $\mathcal{N}(0, 1)$. We use $N = 1000$ samples.

### E.8  Blotto

Blotto is a two-player symmetric zero-sum game, where each player selects a way to place N units onto K fields. The outcome of the game is simply number of fields, where a player has more units than the opponent minus the symmetric quantitiy. We choose N=10, K=5, which creates around 1000 pure strategies, but analogous results were obtained for various other setups we tested. One could ask why is Blotto getting more non-transitive as our strength increases. One simple answer is that the game is permutation invariant, and thus forces optimal strategy to be played uniformly over all possible permutations, which makes the Nash support grow. Real world games, on the other hand, are almost always ordered, sequential, in nature.

### E.9 Kuhn Poker

Kuhn Poker [14] is a two-player, sequential-move, asymmetric game with 12 information states (6 per player). Each player starts the game with 2 chips, antes a single chip to play, then receives a face-down card from a deck of 3 cards. At each information state, each player has the choice of two actions, betting or passing. We use the implementation of this game in the OpenSpiel library [16]. To construct the empirical payoff matrices, we enumerate all possible policies of each player, noting that some of the enumerated policies of player 1 may yield identical outcomes depending on the policy of player 2, as certain information states may not be reachable by player 1 in such situations. Due to the randomness involved in the card deals, we compute the average payoffs using 100 simulations per pair of policy match-ups for players 1 and 2. This yields an asymmetric payoff matrix (due to sequential-move nature of the game), which we then symmetrise to conduct our subsequent analysis.

### E.10 Parity Game of Skill

Let us define a simple $n$-step game (per player), that has game of skill geometry. It is a two-player, fully-observable, turn based game that lasts at most $n$-steps. Game state is a single bit $s$ with initial value 0. At each step, player can choose to: 1) flip the bit ($a_1$); 2) guess that bit is equal to 0 ($a_2$); 3) guess the bit is equal to 1 ($a_3$); 4) keep the bit as it is ($a_4$). At (per player) step $n$ the only legal actions are 2) and 3). If any of these two actions is taken, game ends, and a player wins iff it guessed correctly. Since the game is fully observable, there is no real "guessing" here, agents know exactly what is the state, but we use this construction to be able to study the underlying geometry in the easiest way possible. First, we note that this game is $n - 1$-bit communicative, as at each turn agents can transmit $\log_2(|\{a_1, a_3\}|) = 1$ bits of information, and game lasts for $n$ steps, and the last one cannot be used to transfer information. According to Theorem 1 this means that every antisymmetric payoff of size $2^{n-1 \times n-1}$ can be realised. Figure 7 shows that this game with $n = 3$ has hundreds of cycles, and Nash clusters of size 40, strongly exceeding lower bounds from Theorem 1. Since there are just 161 pure strategies, we do not have to rely on sampling, and we can clearly see Spinning Top like shape in the game profile.

## F  Other games that are not Games of Skill

Table 3 shows a few Noisy Elo Games, which cause Nashes to grow significantly over the transitive dimension. We also run analysis on Kuhn-Poker, with 64 pure policies, which seems to exhibit analogous geometry to Blotto game. Finally, there is also pure Rock Paper Scissor example, with everything degenerating to a single point.

## G  Empirical Game Strategy Sampling

We use OpenSpiel [16] implementations of AlphaBeta and MCTS players as base of our experiments. We expand AlphaBeta player to `MinMax(d, s)`, which runs AlphaBeta algorithm up till depth $d$, and if it did not succeed (game is deeper than $d$) then it executes random action using seed $s$ instead. We also define `MaxMin(d, s)` which acts in exactly same way, but uses flipped payoff (so seeks to lose). We also include `MinMax'(d, s)` and `MinMax(d, s)` which act in the same way as before, but if some branches of the game tree are longer than $d$, then they are assumed to have value of 0 (in other words these use the value function that is contantly equal to 0). Finally we define $\text{MCTS}(k, s)$ which runs $k$ simulations, and randomness is controlled by seed $s$. With these 3 types of players, we create a set of agents to evaluate of form:

- `MinMax(d,s)` for each combination of

$$d \in \{0, 1, 2, 3, 4, 5, 6, 7, 8, 9\}, s \in \{1, \dots, 50\}$$

- `MinMax'(d,s)` for each combination of

$$d \in \{1, 2, 3, 4, 5, 6, 7, 8, 9\}, s \in \{1, \dots, 50\}$$

Table 3: Top row, from left: Noisy Elo games with $\epsilon = 0.1, 0.5, 1.0$ respectively. Middle row, from left: Blotto with $N, K$ equal $5, 3$; $5, 5$, $10, 3$ and $10, 5$ respectively. Bottom row, from left: Kuhn-Poker and Rock Paper Scissors.

- `MaxMin(d,s)` for each combination of

$$d \in \{1, 2, 3, 4, 5, 6, 7, 8, 9\}, s \in \{1, \dots, 50\}$$

- `MaxMin'(d,s)` for each combination of

$$d \in \{1, 2, 3, 4, 5, 6, 7, 8, 9\}, s \in \{1, \dots, 50\}$$

- `MCTS(k,s)` for each combination of

$$k \in \{10, 100, 1000\}, s \in \{1, \dots, 50\}$$

This gives us 2000 pure strategies, that span the transitive axis. Addition of MCTS is motivated by the fact that many of our games are too hard for AlphaBeta with depth 9 to yield strong policies. Also `MinMax(0,s)` is equivalent to a completely random policy with a seed $s$, and thus acts as a sort of a baseline for randomly initialised neural networks. Each of players constructed this way codes a pure strategy (as thanks to seeding that act in a deterministic way).

# H   Empirical Game Payoff computation

For each game and pair of corresponding pure strategies, we play 2 matches, swapping which player goes first. We report payoff which is the average of these two situations, thus effectively we symmetrise games, which are not purely symmetric (due to their turn based nature). After this step, we check if there are any duplicate rows, meaning that two strategies have exactly the same payoff against every other strategy. We remove them from the game, treating this as a side effect of strategy sampling, which does not guarantee uniqueness (e.g. if the game has less than 2000 pure strategies, than naturally we need to sample some multiple times). Consequently each empirical game has a payoff not bigger than $2000 \times 2000$, and on average they are closer to $1000 \times 1000$.

# I   Fitting spinning top profile

For each plot relating mean RPP to size of Nash clusters, we construct a dataset

$$X := \{(x_i, y_i)\}_{i=1}^k = \left\{ \left( \tfrac{1}{k} \sum_{j=1}^k \mathrm{RPP}(\mathrm{C}_i, \mathrm{C}_j), |\mathrm{C}_i| \right) \right\}_{i=1}^k.$$

Next, we use Skewed Normal pdf as a parametric model:

$$\psi(x|\mu, \sigma, \alpha) = \sigma^2[2\phi((x - \mu)/\sigma^2)\Phi(\alpha(x - \mu)/\sigma^2)],$$

where $\phi$ is a pdf of a standard Gaussian, and $\Phi$ its cdf. We further compose this model with simple affine transformation since our targets are not normalised and not guaranteed to equal to 0 in infinities:

$$\psi'(x|\mu, \sigma, \alpha, a, b) = a\psi(x|\mu, \sigma, \alpha) + b, \cdot$$

and find parameters $\mu, \sigma, \alpha, a, b$ minimising

$$\ell(\mu, \sigma, \alpha, a, b) = \sum_{i=1}^k \|\psi'(x_i|\mu, \sigma, \alpha, a, b) - y_i\|^2.$$

In general, using probability of data under the MLE skewed normal distribution model could be used as a measure of "game of skillness", but its applications and analysis is left for future research.

# J   Counting pure strategies

For a given 2 player turn-based game we can compute number of behaviourally different pure strategies by traversing the game tree, and again using a recursive argument. Using notation from previous sections, and $z_j$ to denote number of pure strategies for player $j$ we put, for each state $s$ such that $p(s) = j$:

$$z_j(s) = \begin{cases} 1 & , \text{if terminal}(s) \\ \sum_{s' \in c(s)} \left[ \prod_{s'' \in c(s')} z_j(s'') \right] & , \text{otherwise} \end{cases}$$

where the second equation comes from the fact, that two pure strategies are behaviourally different if there exists a state, that both reach when facing some opponent, and they take different action there. So to count pure strategies, we simply sum over all our actions, but need to take product of opponent actions that follow, as our strategy needs to be defined in each of possible opponent moves, and each such we multiply in how many ways we can follow from there, completing the recursion. If we now ask our strategies to be able to play as both players (since in turn-based games are asymmetric) we simply report $z_1(s_0) \cdot z_{-1}(s_0)$, since each combination of behaviour as first and second player is a different pure strategy.

For Tic-Tac-Toe $z_1(s_0) \approx 10^{124}$ and $z_{-1}(s_0) \approx 10^{443}$ so in total we have approximately $10^{567}$ pure strategies that are behaviourally different. Note, that behavioural difference does not imply difference in terms of payoff, however difference in payoff implies behavioural difference. Consequently this is an upper bound on number of size of the minimal payoff describing Tic-Tac-Toe as a normal form game.

# K  Deterministic strategies and neural network based agents

Even though neural network based agents are technically often mixed strategies in the game theory sense (as they involve stochasticity coming either from Monte Carlo Tree Search, or at least from the use of softmax based parametrisation of the policy), in practise they were found to become almost purely deterministic as training progresses [19], so modelling them as pure strategies has empirical justification. However, study and extension of presented results to the mixed strategies regime is an important future research direction.

# L  Random Games of Skill

We show that random games also exhibit a spinning top geometry and provide a possible model for Games of Skill, which admits more detailed theoretical analysis.

**Definition 4 Random Game of Skill.** *We define a payoff of a Random Game of Skill as a random antisymmetric matrix, where each entry equals:*

$$\mathbf{f}(\pi_i, \pi_j) := \tfrac{1}{2}(Q_{ij} - Q_{ji}) = \tfrac{1}{2}(W_{ij} - W_{ji}) + S_i - S_j$$

*where $Q_{ij} = W_{ij} + S_i - S_j$, and $W_{ij}, S_i$ are iid of $\mathcal{N}(0, \sigma_W^2)$ and $\mathcal{N}(0, \sigma_S^2)$ respectively, where $\sigma = \max\{\sigma_W, \sigma_S\}$.*

The intuition behind this construction is that $S_i$ will capture part of the *transitive strength* of a strategy $\pi_i$. If all the $W_{ij}$ components were removed then the game would be fully *monotonic*. It can be seen as a linear version of a common Elo model [8], where each player is assigned a single ranking, which is used to estimate winning probabilities. On the other hand, $W_{ij}$ is responsible for encoding all interactions that are specific only to $\pi_i$ playing against $\pi_j$, and thus can represent various non-transitive interactions (i.e. cycles) but due to randomness, can also sometimes become transitive.

Let us first show that the above construction indeed yields a Game of Skill, by taking an instance of this game of size $n \times n$.

**Proposition 4.** *If $\max_{i,j} |W_{ij}| < \frac{\alpha}{2}$ then the difference between maximal and minimal $S_i$ in each Nash cluster $C_a$ is bounded by $\alpha$:*

$$\forall_a \max_{\pi_i \in C_a} S_i - \min_{\pi_j \in C_a} S_j \leq \alpha.$$

*Proof.* Let us hypothesise otherwise, so we have a Nash with strategy $\pi_a$ and $\pi_b$ such that $S_a - S_b > \alpha$. Let us show that $\pi_a$ has to achieve better outcome against each strategy $\pi_c$ than $\pi_b$

$$
\begin{aligned}
&\mathbf{f}(\pi_a, \pi_c) - \mathbf{f}(\pi_b, \pi_c) \\
&= \tfrac{1}{2}(W_{ac} - W_{ca} - W_{bc} + W_{cb}) + (S_a - S_b) \\
&> \tfrac{1}{2}(W_{ac} - W_{ca} - W_{bc} + W_{cb}) + \alpha \\
&\geq 0
\end{aligned}
\tag{4}
$$

consequently $\pi_b$ cannot be part of the Nash, contradiction.

Furthermore Nashes supports will be highest around $0$ transitive strength, where most of the probability mass of $S_i$ distribution is centred, and go towards $0$ as they go to $\pm\infty$.  □

First, let us note that as the ratio of $\sigma_S$ to $\sigma_W$ grows, this implies that the number of Nash clusters grows as each of them has upper bounded difference in $S_i$ by $\alpha$ that depends on magnitude of $\sigma_W$, while high value of $\sigma_S$ guarantees that there are strategies $\pi_i$ with big differences in corresponding $S_i$'s. This constitutes of the transitive component of the random game. To see that the clusters sizes are concentrated around zero, lets note that because of the zero-mean assumption of $S_i$, this is where majority of $S_i$'s are sampled from. As a result, there is a higher chance of $W_{ij}$ forming cycles there, then it is in less densely populated regions of $S_i$ scale. With these two properties in place . Figure 8 further visualises this geometry. This shape can also be seen by considering the limiting distribution of mean strengths.

**Proposition 5.** *As the game size grows, for any given $k \in [n]$ the average payoff $\frac{1}{n}\sum_{i=1}^{n}\mathbf{f}(\pi_k, \pi_i)$ behaves like $\mathcal{N}(S_k, \frac{2\sigma^2}{n})$.*

*Proof.*

$$\frac{1}{n}\sum_{j=1}^{n}\mathbf{f}(\pi_k, \pi_j) = S_k + \frac{1}{n}\sum_{j=1}^{n}W_{kj} - \frac{1}{n}\sum_{j=1}^{n}S_j.$$

Using the central limit theorem and the fact that $\mathbb{E}[W_{ij}] = \mathbb{E}[S_j] = 0$, $1 \leq j \leq n$ and that these variables have a variance bounded by $\sigma^2$. $\square$

Now, let us focus our attention on training in such a game, given access to a uniform improvement oracle, which given a set of $m$ opponents returns a uniformly selected strategy from strategy space, among the ones that beat all of the opponents, we will show probability of improving average transitive strength of our population at time $t$, denoted as $\bar{S}_t$.

**Theorem 4.** *Given a uniform improvement oracle we have that, $\bar{S}_{t+1} > \bar{S}_t - W$, where $W$ is a random variable of zero mean and variance $\frac{\sigma^2}{m^4}$. Moreover, we have $\mathbb{E}[\bar{S}_{t+1}] > \mathbb{E}[\bar{S}_t]$.*

*Proof.* Uniform improvement oracle, given a set of index of strategies $I_t \subset [n]$ (the current members of our population) returns an index $i_t$ such that,

$$\forall_{i \in I_t}\mathbf{f}(\pi_{i_t}, \pi_i) > 0 \quad \text{i.e.} \quad \pi_{i_t} \text{ beats any } \pi_i, \ i \in I_t$$

and creates $I_{t+1}$ that consists in replacing a randomly picked $i \in I_t$ by $i_t$. If the oracle cannot return such index then the training process stops. What we care about is the average skill of the population described by $I_t$, $\bar{S}_t := \frac{1}{m}\sum_{i \in I_t} S_i$, where $m := |I_t|$. By the definition of a uniform improvement oracle we have,

$$\forall_{i \in I_t}\mathbf{f}(\pi_{i_t}, \pi_i) > 0 \tag{5}$$

Thus, if we call $a := i_t$ and $b$ is the index of the replaced strategy we get

$$\frac{1}{m}\sum_{i \in I_{t+1}} S_i = \frac{1}{m}\sum_{i \in I_t} S_i + \frac{1}{m}(S_a - S_b) \tag{6}$$

$$= \frac{1}{m}\sum_{i \in I_t} S_i + \frac{1}{m}(Q_{ab} - \tilde{W}_{ab}) \tag{7}$$

$$> \frac{1}{m}\sum_{i \in I_t} S_i - \frac{1}{m}\tilde{W}_{ab}. \tag{8}$$

where $\tilde{W}_{ij} := \frac{1}{2}(W_{ij} - W_{ji})$. This concludes the first part of the theorem. For the second part we notice that since the strategy in $I_t$ is replaced uniformly and $\tilde{W}_{ij}$, $1 \leq i < j \leq n$ are independent of variance bounded by $\sigma^2$, we have,

$$\mathrm{Var}\left[\frac{1}{m}W_{aj}\right] = \frac{1}{m^2}\mathbb{E}\left[\frac{1}{m}\sum_{j \in I_t} W_{aj}\right] = \frac{\sigma^2}{m^4} \tag{9}$$

Finally taking the expectation conditioned on $I_t$, we get

$$\mathbb{E}\left[\frac{1}{m}\sum_{i \in I_{t+1}} S_i | I_t\right] > \frac{1}{m}\sum_{i \in I_t} S_i. \tag{10}$$

$\square$

The theorem shows that the size of the population, against which we are training, has a strong effect on the probability of transitive improvement, as it reduces the variance of $W$ at a quartic rate. This result concludes our analysis of random Games of Skill, we now follow with empirical confirmation of both the geometry and properties predicted made above.

Figure 5: Partial execution of the n-communicativeness algorithm for Tic-Tac-Toe. Black nodes represent states that no longer can reach all possible outcomes. Green ones last states before all the children nodes would be either terminating or are coloured black. The selected children states (building subset $A$) are encoded in green (for crosses) and blue (for circles), with the edge captioned with number of bits transmitted (logarithm of number of possible children), minimum number of bits one can transmit afterwards, and minimum number of bits for the other player (because it is a turn based game). $n$ at each node is minimum of $\phi_x$ and $\phi_o$, while for a player $p$ making a move in state $s$, we have $\phi_p = \max_{A \subset c(s)}\{\log_2 |A| + \min_{s' \in A} \min_{p'} \phi_{p'}(s')\}$. Red states are the one not selected in the parent node by maximisation over subsets.

Figure 6: Visualisation of construction from Proposition 1. Left) split of the 19 x 19 Go board into regions where black stones (red), and white stones (blue) will play. Each player has 180 possible moves. Centre) Exemplary first 7 moves, intuitively, ordering of stones encodes a permutation over 180, which corresponds to $\sum_{i=1}^{180} \log_2(i) = \log_2(\prod_{i=1}^{180} i) = \log_2(180!) \approx 1000$ bits being transmitted. Right) After exactly 360 moves, board will always look like this, at which point depending on $\mathbf{P}_{\mathrm{id(black),id(white)}}$ black player will resign (if it is supposed to lose), or play the centre stone (if it is supposed to win).

Figure 7: Game profile of Parity Game of Skill with 3 steps. Note that its Nash clusters are of size 40, and number of cycles exceeds 140, despite being only 2-bit communicative.

Figure 8: Game profile of the random Game of Skill. Upper left: payoff matrix; Upper right: relation between fraction of strategies beaten for each strategy and number of RPS cycles it belongs to (colour shows which Nash cluster this strategy belongs to); Lower left: payoff between Nash clusters in terms of RPP [2]; Lower right: relation between fraction of clusters beaten wrt. RPP and the size of each Nash cluster. Payoffs are sorted for easier visual inspection.