[Reviews · NeurIPS 2020]

Review 1

Summary and Contributions: This paper investigates the transitivity of deterministic policies in what they call "real world games", e.g. board games, video games, etc. as opposed to the game theory sense ("multi-agent strategic interactions"). They hypothesize that "games of skill" (a) can be decomposed into a transitive hierarchy of non-transitive classes of strategies, and (b) the "best" and "worst" classes are much smaller than the classes in the middle. The authors prove a lower bound on the maximum cycle size for EFGs. The authors then show that a fixed-memory variant of fictitious play converges in a "game of skill", and that you can separate a set of strategies into these transitive layers by Nash clustering (a prior technique), and that training via self-play on only the strategies in the "best" layer is sufficient for improvement in self-play. Finally, the authors look at the empirical size of these layers (with a bunch of approximations) in some OpenSpiel games, and find that most of them have this "bell curve" distribution of layer sizes, and show that PBT has a "phase change" in population size; once it's large enough, it stops cycling.

Strengths: Novelty --------- This paper takes a creative new approach to analyzing the structure of games, following but extending a line of prior work on transitive/cyclical strategies in games [1]. This analysis provides interesting intuitions for understanding self-play techniques such as PBT; in particular, I find the following claim quite fascinating: "the complexity of the non-transitivity discovery/handling methodology decreases as the overall transitive strength of the population grows. Various agent priors (e.g. search, architectural choices for parametric models such as neural networks, smart initialisation such as imitation learning etc.) will initialise in higher parts of the spinning top, and also restrict the set of representable strategies to the transitively stronger ones. This means that there exists a form of balance between priors one builds into an AI system and the amount of required multi-agent learning complexity required (see Figure 4 for a comparison of various recent state of the art AI systems). It would be interesting to see this claim empirically validated, although that's probably beyond the scope of this work. Soundness of theorems and empirical evaluation -------------------------------------------------------- The theorems are clear and I believe correct (I didn't read the proofs because they are pretty intuitive and almost trivial), and the experimental evaluation is extensive. In particular I like the demonstration in Table 1 (RHS) that there really is a fixed population size above which cyclic behavior stops occurring [1] David Balduzzi, Marta Garnelo, Yoram Bachrach, Wojciech M Czarnecki, Julien Perolat, Max Jaderberg, and Thore Graepel. Open-ended learning in symmetric zero-sum games. ICML, 2019.

Weaknesses: Significance: —————— In terms of the motivation for this paper, I was surprised by: "Interestingly, we show that many games from classical game theory are not Games of Skill, and as such might provide challenges that are not necessarily relevant to developing AI methods for real world games." I was always under the impression that board games etc. were toy problems that practitioners in our field were using as proxies for real-world strategic interaction, not the other way around! If anything, game theory abstracts away the "board game" elements and deals directly with stylized strategic interactions (bargaining, social dilemmas, etc.) If the authors want to focus specifically on "games" themselves, I suppose that's fair, but I would encourage the authors to explore the converse statement: "We discover idiosyncratic characteristics of "games of skill", which perhaps suggests that our field's preoccupation with these games may not transfer well to real-world strategic interactions." Soundness of the claims ———————————— The claim in the Intro that the spinning-top geometry has to do with “what makes games interesting”, seems undermined by Theorem 1 which I believe shows that almost any EFG will have long cycles. Intuitively, the Theorem 1 construction basically creates a cycle of strategies by players communicating in the first half who is supposed to win, and then using the second half for one player to “throw the game”. You can do this in an EFG with a random payoff matrix, can’t you? ** Now, perhaps the salient part of the “spinning top” is the transitive structure. But no theory is provided about the transitive structure or the cycle size at different levels. (I think you *could* prove a limit on the cycle size based on the support of the NE). Table 1 provides some empirical evidence about this structure, but again: wouldn’t a random matrix of winners also have a binomial distribution of “Fraction beaten”, leading to this same bell curve shape? ** Despite this criticism, I do feel that there’s a good intuition in Theorem 1, because it actually does describe how RL would behave in terms of deterministic strategies (each iteration would only learn to win along the path of play of their deterministic opponent) and demonstrates how FP could need an exponentially large buffer in EFGs. However, if strategies are not deterministic and instead have support on all actions, then a BR to that strategy should *always* be a transitive improvement in a perfect-info game, correct? This difference between perfect info and imperfect info games seems relevant for self-play/PBT, and perhaps salient to the discussion in Sec 4. Nit —— I’m not sure if I’m understanding correctly, but the conditions in Def 1 and Theorem 1 seem redundant. The condition “before reaching the node whereafter at least one of the outcomes…” in Def 1 guarantees that P is such that both wins and losses follow from each of the 2^n “messages”, so Thm 1 could just say “there exists a cycle of 2^n pure strategies”, with no further restrictions on P, right?

Correctness: Yes

Clarity: Yes, it's very well written, prose is clear, diagrams are excellent, and supplement is extensive.

Relation to Prior Work: Yes

Reproducibility: Yes

Additional Feedback: Overall, I found this work creative and interesting to read. However, between the focus on recreational games and my skepticism about how meaningful this “spinning top structure” is, I’m left wondering what significance these observations have to the field. I would be very interested to hear more specific thoughts in the rebuttal on how these observations should inform future work on learning in games. --------------------------------------------------------------------------- Response to Author Feedback: Thanks for the response! First off, in the past few weeks I’ve noticed that in my own reading and research, this work has provided a novel perspective that suggests new ways of approaching MARL problems. So even if I’m not convinced of every conclusion in this work, I think it could provide value to the community by sparking new ideas. Re: “…board games are not considered toy problems…” Exactly! This did not come across to me in the manuscript. I think of game theory as encoding (stylized) “general strategic interactions”, so I read the claims in the text as “these results apply to board games, rather than general strategic interactions”. If you want to say “*real* temporally-extended strategic interaction (as modeled by board games) have a different geometry than their stylized matrix-game counterparts”, perhaps conjecture that more explicitly, or at least address this issue. (Note even in the Author Response you use the GoS definition of “games created for human’s enjoyment”) Re Appendix L (not J), this section is nice and I missed it, thanks! I see this Appendix as showing (1) “G has a transitive and circular component” —> (2) “strategies in G have a Gaussian (“spinning top”) geometry”. I would like to know whether random (deep) EFGs have property (1) ! The counterexamples of RPS (not an EFG) and Kuhn (depth 2) don’t count. I think the following additional experiments would clarify the work substantially: 1. Analyze how cycle length grows with size in matrix games and EFGs. Does max cycle length grow exponentially with depth in EFGs: 2. Empirically examine whether random MGs and random EFGs have a spinning top geometry.


Review 2

Summary and Contributions: This paper examines the geometric structure of the space of strategies for real-world games. The paper proposes a hypothesis, called the Game of Skill hypothesis, which states that these geometries will look like a spinning top for real-world games, that is, games played by people where there are enough possible strategies to keep things interesting and where-in players can improve over time as they learn and play the game and gain more skill. The paper then proceeds to demonstrate theoretically that a large set of games has this structure, and finally experimentally show that this structure exists in many real-world games.

Strengths: This paper has a clear hypothesis and rigorous demonstration of that hypothesis both theoretically and empirically. This is to my knowledge a novel hypothesis and observation which, while it might not change current practices, certainly offers an underlying justification and explanation of when certain training methods work and when others don't, or when they might be unnecessary. I think that this can be a significant contribution and have an impact on how researchers and practitioners think about the game domains they address. It is also a hypothesis with a certain beauty to it. It has great relevance to the NeurIPS community, as many in that community are concerned with learning in these types of games.

Weaknesses: No high level weaknesses were noted. Some minor concerns are discussed below.

Correctness: The theoretical claims and empirical methodologies are sound and appear correct.

Clarity: The paper is very well written. It was very enjoyable to read.

Relation to Prior Work: This is perhaps one area that could be improved, but I am not sure how much previous work has specifically looked at the geometric structure of games. The authors do a good job of positioning the work in the context of successful trained agents in a wide variety of domains, but it would also be interesting for there to be a discussion of any other attempts to understand the strategy space of games in a general way, if such previous attempts exist.

Reproducibility: Yes

Additional Feedback: What assumptions did you make about the possible action frequency of the games in proposition 2 to get that lower bound? Are there any metrics that you can use to state how good of fits the lines in Table 1 are? Some of them (Quoridor 4x4) look like really tight fits, while others (Alphastar) seem like not very good fits. Does this just depend on how the size of the Nash clusters change as the we increase the layers? ----------------------------------------- Response to Author Feedback ----------------------------------------- I think the authors provided adequate feedback to the concerns of Reviewer 1, and answered a few of my questions as well. I think that this paper provides a valuable new perspective that could spark novel ideas and approaches in these important areas.


Review 3

Summary and Contributions: This paper presents the spinning-top hypothesis, which states that real world games of skill have a certain geometry when plotting win rates against the number of different strategies that dominate other strategies when topologically sorted. The paper proves a number of theorems about the size of cycles of transitive strategies and provides empirical evidence supporting the proofs. The paper argues that knowing that skill games have a top-shaped geography is useful in designing multi-agent training regimes, but does not provide any support for this claim.

Strengths: theoretical grounding: the paper is a theory paper, and presents what appear to be correct proofs in the supplemental materials. empirical evaluation: n/a significance: The paper may in fact provide insights to the training of game-playing agents. The significance is not immediately obvious. This may be the sort of paper where the significance is not clear until other people do follow on work. novelty: The paper appears novel. There is some evidence that similar analyses are conducted in other domains, but this is the first to this reviewer's knowledge looking at games. The insights about skill vs non-skill games are fairly intuitive, but the paper does provide some mathematical grounding. relevance to the NeurIPS community: Unclear.

Weaknesses: significance: As the paper currently stands, the significance of the paper is low in the sense that practitioners of game-playing ML would hope to see some practical suggestions on how the hypotheses and theorems will improve the ability to effectively and efficiently train game playing agents. The suggestions are not made, but only teased at. The author's claims that the spinning top hypothesis may help are unsubstantiated at this time. Were the authors to show that a training regime could be changed based on insights from this paper. relevance to the NeurIPS community: The paper will be relevant to a portion of the NeurIPS community, though a large amount of the community is focused on practical aspects of machine learning.

Correctness: The paper appears correct. The insights about skill vs non-skill games are not surprising to this reviewer, who has experience in AI game playing and AI game design. We should thus expect to see theorems such as those presented and the proofs look reasonable.

Clarity: The paper lacks clarity in the introduction. The introduction makes heavy use of game theoretic terminology, which often overloads terminology more common to the game agent ML community. The terminology is mostly cleared up in the preliminaries section, but some terms and definitions are hard to follow throughout the paper. Examples (e.g. rock-paper-scissors) tend to make the paper easier to understand when used, but they are not used very often.

Relation to Prior Work: Related work seems reasonable, though outside the scope of this reviewer's background. Potentially of use: a paper by Santiago Ontanon (or David Churchill) gives the branching factor of StarCraft. However, this detail is not particularly relevant other than it provides evidence for why we would see large numbers of strategies for real world games.

Reproducibility: Yes

Additional Feedback: The concept of transitivity (e.g., "a strong transitive component of the underlying game structure") is confusing at first. In particular, why is play style (e.g., a preference for certain strategies) a non-transitive component? Perhaps this is terminology from game theory that is not prevalent in the broader NeurIPS community. Indeed, transitivity in AI game playing is usually associated with state transitions in the game, which can lead to some confusion as these readers would be interested in the paper. The usage of "transitivity" only becomes clear on page 4. However, the notion of transitive strength remains unclear. In particular, it is unclear how something can be highly transitive or partially transitive. Does this have to do with the length of cycles? It is challenging to make the jump from f(pi_i, pi_j) having to do with one action to f(policy_i, policy_j). Does a long transitive cycle mean that a single policy will eventually beat another policy? Does it mean that in a game one will get to a point in a game where many actions pi_i 0...N are available and that there is a cycle created when considering which actions will result in winning outcomes e.g. i>j>k>i ? See Ontanon for an analysis of the branching factor of StarCraft's extended form. Why would we want to build agents that lose to some opponents in multi-agent training regimes? Obviously it will fail of a game were non-transitive. Perhaps an example of a non-transitive game (a game without a cycle?) would help to understand. This reviewer is thinking of games that are based purely (or highly?) on randomness are non-transitive. In this case, it would make sense that a multi-agent training regime would fail because the outcome of the game would not be a function of the choices the agent makes--it's policy. Is it correct to say that all games of skill are transitive and all games of chance are non-transitive? Is a game style the same as a policy? What is the relationship between number of simulations in MCTS (and search depth in alpha-beta) and game style, or skill? One might argue that number of simulations or search depth equates to skill level, but the comparison to human skills is not quite that clean. That is, a skilled chess player may not actually perform deeper search than a novice. It is unclear the value of having agents that are designed to lose. It allows investigation of the lower part of the cone, but given that no agents would ever be designed in the real world to lose, is it not unreasonable that there is just a naturally occurring lower threshold. Certainly that means the geometry would not look like a spinning top anymore, which is aesthetically not pleasing, but perhaps more realistic. On the other hand, it could be argued that an agent designed to lose is just an agent that is extremely unskilled. However, a random agent would be a better way of capturing this, since for some games it may require effort to lose more than chance would suggest (another way of finding a naturally occurring bottom threshold). The paper suggests that the spinning top hypothesis may provide insights that help in designing training multi-agent training regimes for games. However, the paper doesn't deliver any concrete implications. In some ways, the results of the paper are intuitive. We would expect a small number of strategies to yield the highest win rates and a small number of strategies to be so profoundly bad as to almost always lose, and a larger number of strategies to result middle-of-the-road win/loss rates. We expect players with greater skill to win more games than those with less skill. We expect a MCTS agent that does more simulations to win more games than an MCTS agent that performs fewer simulations. How exactly does this help us train agents better or faster? Does this imply that we need more diversity in our population of multi-agents in the middle and that we can prune agents as we reach higher skill levels? Tournaments do this naturally. It is hard to imagine a skill game that has a non-spinning-top shape. What would a game be that had an hour-glass shape? or a spinning top with a pinch in the middle of the bulge? I suppose one could design a theoretical game that created these shapes--a game where there are many ways to be skilled or very poor, but not many ways to be mediocre? Furthermore, the empirical game profiles for skill games are very different. We see some games with the bulges low and high. Is the implication that we should train our agents differently when we see the bulge high or low? It seems the more likely implication will be that this might mean we should expect to see learning progression slow down at some point because it is not clear which agent strategies/policies should be promoted. But in the end, we don't care if there are a lot of mediocre or bad strategies, because we generally want one (or a small number) of agents that dominate all others.

[Author Response · NeurIPS 2020]

First, we would like to thank **R1**, **R2** and **R3** for their evaluation, criticism and detailed comments which will help us improve the quality of our paper. We were pleased that **R1** and **R2** found the paper "very well written" and enjoyed the reading. We also are glad that **R1** and **R2** appreciated the "creativity" and the "beauty" of our hypothesis. We would finally like to thank **R1**, **R2** and **R3** for the mentioned related works and textual corrections; we will make sure to include all of these.

**Answer to R1**. Thanks for the great questions and suggestions. As **R1** points out – our paper indeed focuses purely on "games", not on general "strategic interactions". Exactly because of this restriction, we are able to define and reason about a specific subclass of problems, that have interesting properties we investigate. Note though that it is important to understand that board games are not considered toy problems in the multi-agent learning and agents research community. They pose big challenges for machine learning algorithms, cfr. recent result in for example Go and Chess. They are effectively interesting abstractions of real-world situations, which are still complex but can be studied in a controlled setting. Games from Game Theory on the other hand are often focusing on very distilled, single high level challenge (e.g. RPS, social dilemmas), that abstract away a lot of dynamics. They are typically the first benchmarks to try out new multi-agent RL algorithms. We focus on providing a better understanding, and tools, for those working on what we describe as "real world games" (e.g. StarCraft, DOTA, Quake, Soccer, Go, etc.). We will add a statement explicitly stating the converse (that these properties will not necessarily transfer outside of this scope) in the camera ready version of the paper.

While Theorem 1 indeed shows that many games in EFG (e.g. Appendix L: "Random Games of Skill") will have extremely long cycles, this is not what constitutes the spinning top geometry. This is only one of the properties, that a spinning top like game would have - extremely long cycles around a very weak transitive component. However, many games do not follow this structure (e.g. Kuhn Poker, RPS etc.), thus clearly it is a non trivial class of games. We indeed claim that interesting, strategically complex, competitive, zero-sum, 2-player games imply this sort of geometry, but not the opposite implication (that only interesting games have this geometry).

To summarise, the main lessons learned, that can guide future research are as follows: we identify a set of geometrical properties, that real world games (defined as games created for human's enjoyment, in the form of competitive, strategically deep 2 player games), seem to share. While we were not able to prove theoretically the exact shape (and thus call it a spinning top *hypothesis*), we took a path from other empirical sciences - to form a hypothesis, and even if it is not directly provable - try to list various predictions it would make, and prove them instead: existence of very long cycles around the low transitive score, population size relation to convergence (that was observed in practice in research focusing on such games), and with empirical probing - we were able to see this shape emerging, while lacking in some other games created in game theory literature (thus confirming that the definition is not degenerate). On the theoretical side, we believe we pose a challenging open question of further structuring and formalising geometries of these types of games which, while of interest to the scientific community, so far have been often treated purely as "2 player symmetric zero-sum games". On the practical side, this research provides insights into population-based learning dynamics, required sizes of populations (which are much higher than those currently employed in practical applications), and suggests that in order to tackle these non-transitive challenges one might need more structured, latent policies, where a single set of weights can represent exponentially many types of behaviour (e.g. AlphaStar league "z conditioned agent"). Finally, we believe that these insights allow to construct strategically deep environments for AI to train on, as part of the current efforts of Open Ended Learning, and environment-agents co-evolution efforts. Ensuring the geometry of a spinning top (by creating cyclic interactions between actions on per time stamp basis while preserving skill dimension) can help AI researchers who are bottlenecked by not deep enough problems.

**Answer to R2**. First, we would like to thank the reviewer for the very positive feedback and great suggestions. In terms of proposition 2 we assumed 15hz control (used in e.g. Quake III Capture the Flag project). We could indeed provide bounds for other games, however for board games the bounding is a bit harder (since each move affects legality of future moves), thus Proposition 1 uses a subset of legal moves to provide the lower bound. The overall construction is algorithmicised and provided in the Appendix, and relies on being able to traverse the game tree, potentially with some heuristic choice of subset of actions in each node. We aim to provide bounds for other games as well in the camera ready version. For 3x3 board games we should be able to compute exact $n$.

**Answer to R3**. While we fully agree that there is a huge value in papers providing new algorithmic solutions for game playing agents, we also argue that the significance of understanding the underlying scientific phenomena of learning is equally important. Our paper situates itself exactly in this spectrum, where we aim to provide a better understanding and uncover a previously unknown geometry of the type of games we investigate.

We fully agree with the reviewer, that the problem we are tackling lies exactly in the intersection of classical RL, Game Theory, and Game Design. Unfortunately, all these three communities have independently built different naming conventions for the same objects. Consequently we spent a lot of time trying to unify, and present everything in a way that is possible to parse independently from the background.

[Meta-Review · NeurIPS 2020]

This is a borderline paper. After the rebuttal and discussions, R1 has turned more positive about the paper, while R3 still thinks the paper is not ready for publication at NeurIPS. The paper studies the strategy space of real-world games from an interesting and novel perspective. That said, while the paper teases at how the results might produce more concrete impacts, it is quite hard to predict whether any of the follow-up directions would be fruitful at all, and it would be better if the authors take one of the possible directions and showcase the usefulness of the findings in this paper. After discussions among the reviewers and the meta-reviewers, we decide to accept the paper given that its novel perspectives may spark interesting discussions and new ideas in the community.